# Enhancing Interfacial Bonding and Tensile Strength in CNT-Cu Composites by a Synergetic Method of Spraying Pyrolysis and Flake Powder Metallurgy

**DOI:** 10.3390/ma12040670

**Published:** 2019-02-23

**Authors:** Xiangyang Chen, Rui Bao, Jianhong Yi, Dong Fang, Jingmei Tao, Yichun Liu

**Affiliations:** Faculty of Materials Science and Engineering, Kunming University of Science and Technology, Kunming 650093, China; 13577124031@163.com (X.C.); fangdong@kmust.edu.cn (D.F.); kiwimaya@126.com (J.T.); liuyichun@kmust.edu.cn (Y.L.)

**Keywords:** carbon nanotubes (CNTs), copper composites, interfacial bonding, mechanical properties

## Abstract

Carbon nanotube (CNT)-reinforced metal matrix composites (MMCs) face the problems of dispersion and interfacial wetting with regard to the matrix. A synergetic method of spray pyrolysis (SP) and flake powder metallurgy (FPM) is used in this paper to improve the dispersibility and interfacial bonding of CNTs in a Cu matrix. The results of the interface characterization show interface oxygen atoms (in the form of Cu_2_O) and a high density of dislocation areas, which is beneficial for interfacial bonding. The tensile results show that the tensile strength of the SP-CNT-Cu composites is much higher than that of the CNT-Cu composites when the mass fraction of the CNTs does not reach the critical value. This can be explained by the nanoparticles which are found on the surface of the CNTs during the SP process. These nanoparticles not only increase the tensile strength of the SP-CNT-Cu composites but also improve the dispersion of the CNTs in the Cu matrix. Thereby, uniform dispersion of CNTs, interfacial bonding between CNTs and the Cu matrix, and the enhancement of tensile strength are achieved simultaneously by the synergetic method.

## 1. Introduction

Copper (Cu) is widely employed in electrical and mechanical fields because of its excellent electrical [1], thermal conductivity [2], antifriction [3] and abrasion resistance [4]. However, the lower mechanical strength of Cu matrices is an intrinsic fault, which limits their applications in mechanical aspects. To obtain exceptional mechanical properties for a Cu matrix, introducing carbon nanotubes (CNTs) is one of the effective means. CNTs show excellent strength (up to ~100 GPa), superior elastic modulus (~1 TPa), and a high aspect ratio (>1000) [5,6,7], like other carbon nanomaterials, such as graphene oxide [8] and carbon nanofibers [9]. Therefore, Cu-based composites with CNT reinforcements have received tremendous attention. In order to prepare high-strength Cu-CNT composites, two key issues must be addressed: the dispersion of CNTs in the Cu matrix and the interfacial bonding between CNTs and the Cu matrix.

To solve the dispersion problem of CNTs, various methods have been developed in the past few decades. The common method is the acidification of CNTs. Chen et al. [10] acidified CNTs with vitriol/nitric acid, which cuts the length of CNTs and forms a large number of defects and functional groups on the surface, in order to overcome the van der Waals forces. In addition, Song et al. [11] dispersed 3.0 vol % multi-walled CNTs in Cu powder by high energy ball milling (HEBM). Although the dispersion of CNTs was achieved, HEBM led to the breakage of CNTs during the process. In general, acidification and HEBM can reduce the agglomeration of CNTs, but also gives rise to damage to the CNTs. Fan et al. [12] introduced sodium dodecyl sulfate (SDS) as a dispersing agent to disperse the CNTs. However, these organics might lead to voids or pores in the sintered bulk and early initiation of cracks.

In order to improve the interfacial bonding strength between CNTs and a metal matrix, Chu et al. [13] use Ti as an alloying element to form intermetallic compounds to enhance the interfacial bonding in CNT-Cu-Ti composites. This compound enhances interface bonding by its chemical reaction at the interface. Nevertheless, this intermetallic compound requires CNTs as a carbon source, which can destroy the intact microstructures and excellent mechanical properties of CNTs. It is also a good method for improving the wettability of CNTs and a Cu matrix by electroless deposition of Cu on the surface of CNTs. Daoush et al. [14] adopted the method of electroless deposition, which has achieved good results in both the dispersion of CNTs and their interface bonding with the Cu matrix, and improved the mechanical properties of the CNT-Cu composites. However, this experimental process involves purification, sensitization, and activation [15], which is already too complicated and has a lower repeatability. In view of the shortcomings of the above methods, we were inspired to investigate the use of spray pyrolysis (SP) in preparation of CNT-Cu composite powder. The nanoparticles on the surface of CNTs and copper-containing compounds are formed, not only to improve the dispersion of CNTs, but also to improve interfacial bonding with a Cu matrix [16]. When the CNT-reinforced composites are stretched, a stronger interfacial bonding provides higher load transfer efficiency, increasing the tensile strength of the composite [17].

In this work, flake powder metallurgy (FPM) [18,19] was introduced, because disc-shaped Cu flakes have a good geometric compatibility with CNTs. Spark plasma sintering (SPS) was utilized for densification of the SP-CNT-Cu composite powders, because of the short sintering cycle. In order to clarify the effects of dispersibility of the CNTs and tensile strength of the composite, unreinforced Cu bulk and CNT-Cu composites, as controls, were prepared by FPM only. 

## 2. Experiment

### 2.1. Preparation of CNT-Cu Oxide Composite Powder by Spraying Pyrolysis (SP)

CNT-Cu oxide composite powder from the CNT dispersion solution (CNTs: inner diameter: 5–15 nm, outer diameter: 30–50 nm, length: 2–3 μm, CNT Content: 10 wt % composition; Chengdu Organic Chemistry Co. Ltd., Chengdu, China) and the precursors of Cu(CH_3_COO)_2_·H_2_O (CuAc, 99%, Shanghai Aladdin Biochemical Technology Co., Shanghai, China), were prepared by using an ultrasonic spraying pyrolysis system. CuAc weighing 6.0 g was dissolved in 3.0 L deionized water, and stirred at room temperature for 15 min. CNT dispersion solution weighing 9.0 g (the CNTs’ weight was 0.9 g) was added to the above CuAc solution, and stirred another 15 min. The homogeneous CNT-CuAc precursor solution was transferred to an ultrasonic nebulizer (Beijing Rongsheng Biochemical Technology Co., Beijing, China), and then atomized into small droplets (atomization rate was 3.0 ml/min). Meanwhile, an airflow in a controlled flow rate (airflow speed was 3.5 mL/s) carried the droplets into the heated tubular reactor (the pyrolysis temperature was 700 °C). The droplets were converted into the CNT-Cu oxide composite powder, which was gathered by a collecting installation.

### 2.2. Preparation of CNT-Cu Composites

The obtained CNT-Cu oxide composite powder was heated in flowing N_2_ + H_2_ atmosphere at 300 °C for 3 h to remove the dispersing agent, which contained the CNTs’ dispersion and reduced the CNTs-Cu oxide composite powder. It should be noted that the mass fraction (wt %) of carbon in fabricated CNT-Cu composite powder was 15.89%, as detected in carbon sulfur analyzer. Electrolytic Cu powder (99.9%, Shanghai Naiou Nano Technology Co. Ltd., Shanghai, China) was first milled into flaky Cu power by ball milling in ethanol, at a rotational speed of 300 rpm for 20 h with a ball-to-powder ratio of 10:1. Then, a CNT-Cu composite powder weighing 0.75 g was mixed with 40 g flaky Cu powder by planetary ball milling in ethanol at a rotational speed of 180 rpm for 3 h with a ball-to-powder ratio of 10:1. The mixture was washed and dried under vacuum at 60 °C for 6 h. The mass fraction of CNTs in the composite powder was 0.3% (denoted as SP-0.3 wt % CNT-Cu composite powder). The preparation is as illustrated in Figure 1. In this research, SP-0.1 wt % and SP-0.5 wt % CNT-Cu composite powder samples were fabricated under the same conditions.

Another four groups of composite powder were fabricated as control groups. These control groups (denoted as unreinforced Cu powder, 0.1 wt %, 0.3 wt %, and 0.5 wt % CNT-Cu composite powder respectively) were prepared directly using flaky Cu and CNT dispersion solution in a planetary ball mill, with the above powder blending parameters.

Bulk samples of these composite powders were sintered on a SPS system (SPS-Labox-650, Japan) at 750 °C for 5 min in holding time, under an axial pressure of 50 MPa from the start to the end of the sintering process. The size of the bulk samples was 30 mm in diameter and 3 mm in thickness.

### 2.3. Characterization Tests

The morphology of the starting materials, the CNT-Cu composite powder, the fracture surface, and the interface microstructure of the CNT-Cu composites were observed using field-emission scanning electron microscopy (FE-SEM, Nano Nova 45, Hillsboro, OR, USA) and transmission electron microscopy (TEM, Tecnai G2 TF30 S-Twin, FEI, Hillsboro, OR, USA). TEM samples of the CNT-Cu composites were polished and thinned in an ion milling system (GATAN PIPS, Model 691). The prepared composite powders underwent elemental analysis for determining their carbon content by a Carbon/Sulfur analyzer (ELTRA CS800, GmbH, Heppenheim, Germany). The phases of composite powders were identified using an X-ray diffractometer (XRD, D/Max 2400, Tokyo, Japan), with a Cu-K_α_ radiation source (*λ* = 1.54056Å, with an operating voltage of 40 kV and current of 30 mA), and an image plate detector covering the 2θ range 20°–80°. The relative density of bulk was measured by the Archimedes method. A hardness test was carried out using a Vickers tester (MC010, Yanrun Company, Shanghai, China) at a load of 0.98 N, with a dwell time of 15 s. The tensile samples were machined to dog bone-shaped specimens with a gauge length of 15 mm, width of 2 mm, and thickness of 1.6 mm. A tensile test was conducted on a Shimadzu AG-IS5KN Tensile Tester, with a crosshead speed of 0.2 mm/min at room temperature.

## 3. Results and Discussion

### 3.1. Starting Materials

The morphologies and structures of starting materials have been investigated, as shown in Figure 2. Figure 2a shows the typical morphology of the electrolytic Cu powder. After ball milling, all electrolytic Cu powders are transformed into flaky shapes, as shown in Figure 2b. The flaky Cu powder has two relatively flat planes compared to the dendritic electrolytic Cu powder, which has a better geometric compatibility with the CNTs. This is an advantage for the CNTs to adhere to the surface of the flaky powder. 

Due to the Van der Waals forces between the CNTs themselves, the CNTs tend to agglomerate together. In order to facilitate the dispersion and removal of the catalyst remaining on the surface of the CNTs, an acidification method is usually employed. However, the structural integrity of the CNTs can be destroyed during the acidification treatment, thus introducing an adverse influence on their mechanical properties. In this study, as shown in Figure 2c, no obvious entanglement has been observed, and the original nanostructure keeps its integrity in the CNT dispersion. In addition, CNTs which are not significantly knotted and entangled were more advantageous for the subsequent modification by SP process.

### 3.2. Composite Powders

For the purpose of examining the phase composition of the composite powder, XRD measurements were carried out. The black curve of the spectrum in Figure 3 shows the XRD spectrum of the CNT-Cu oxide composite powder prepared by SP. The main diffraction peaks are Cu_2_O and CuO, indicating that a chemical reaction occurred during the SP process. It can be observed from the black curve of the spectrum that the CuAc is decomposed into Cu_2_O and CuO, which seems to indicate that the decomposition of CuAc is divided into multiple processes. Mansour et al. [20] have also revealed similar thermal decomposition results. After the CNT-Cu oxide composite powder is reduced, the red curve of XRD spectrum reveals that the characteristic peaks of Cu are the most obvious, but that there are still distinguishing peaks of Cu_2_O with extremely low diffraction peak intensity. This is due to the chemically bonded oxygen-containing functional groups on the CNTs, which are not reduced easily, so even Cu_2_O and CuO have almost been reduced to Cu. Therefore, under the reducing conditions of this paper, there is still a very small amount of Cu_2_O. These oxygen atoms would play an important role in the interface of the CNTs and the matrix.

Figure 4a shows the microscopic morphology of the CNT-Cu composite powder. It reveals that the CNTs are effectively dispersed, and some of them are single dispersed. Copper nanoparticles are attached to the outer wall of the CNTs. Therefore, the CNT-Cu composite powder can be uniformly dispersed in the Cu powder matrix, due to its relatively decentralized structure by a moderate ball milling process. It is beneficial to retain the structure in which the nanoparticles attached to CNTs and to maintain the integrity of the CNTs. TEM and high resolution transmission electron microscopy (HRTEM) are used to investigate the interfacial structure between CNTs and nanoparticles. Nanoparticles with sizes ranging from 20–50 nm are uniformly attached to the surface of the CNTs, as shown Figure 4b, and Figure 4c shows that the copper nanoparticles are attached to the CNTs. These nanoparticles act to modify the CNTs, which is beneficial for improving the dispersion of CNTs in the copper powder matrix. Furthermore, this structure of copper particles attached to CNTs could be reserved after low-energy ball milling. The formation of a coating layer [21,22] or nanoparticles [23] by chemical reaction in the surface of the CNTs is advantageous for improving the dispersibility of the CNTs and their wettability with regard to the metal matrix. Hence, the copper nanoparticles as shown in Figure 4 can enhance interfacial bonding effectively and improve the load transfer capability from CNTs to the metal matrix.

To verify the uniform dispersion of the CNTs in the Cu matrix, SEM images of the composite powder samples with various contents of CNTs are exhibited in Figure 5. It can be seen from Figure 5a–d that some of the CNTs are uniformly embedded in the flaky Cu powder, while others are attached to the surface of the flaky Cu. Moreover, when the mass fraction of CNTs is less than 0.5%, no significant agglomeration of CNTs is observed. This suggests that the CNTs can be dispersed effectively in flaky Cu powder. This could be attributed to the fact that: (1) the plane of the flaky Cu powder is favorable for the attaching of CNTs, (2) during the ball milling, the CNTs can effectively embed into the matrix. However, when the mass fraction of CNTs is increased to 0.5%, although no significant CNTs agglomeration occurs, CNTs bundles are observed in the flaky Cu powder, (Figure 5e,f). This has a negative impact on the mechanical properties of the composites. By comparison, we found that the dispersion of CNTs in SP-CNT-Cu is better than that of the CNT-Cu composite powder. It further shows that SP is advantageous for improving the dispersibility of CNTs in the metal matrix.

### 3.3. Tensile Property and Fracture Morphology

Tensile stress-strain curves of unreinforced Cu, CNT-Cu composites, and SP-CNT-Cu composite samples are shown in Figure 6. The unreinforced Cu bulk sample exhibits an ultimate tensile strength (UTS) of 234.2 MPa and an elongation of 43.3%. The 0.5 wt % CNT-Cu composite is too brittle to get strain hardened, which shows a UTS of 274.1 MPa and ductility of 2.9%. While for the 0.1 wt % and 0.3 wt % CNT-Cu composites, strength enhancement is achieved, which show UTS of 261.2 MPa and 306.2 MPa, and ductilities of 14.6% and 11.7%, respectively. Compared to the unreinforced Cu sample, the UTS are improved by 11.5% and 30.7%, respectively. It can be seen from Figure 6 that the SP-0.5 wt % CNT-Cu composite is brittle and the elongation is relatively low (only 7.6%). The UTS of SP-0.1 wt % and SP-0.3 wt % CNT-Cu composites are enhanced to 295.5 MPa and 353.9 MPa, respectively. However, the strength of these composites is enhanced by CNTs, and their ductility is significantly reduced with the mass fraction of CNTs. A similar situation has also been reported [24,25,26,27]. It has been the research target to increase the strength and ductility of the composite at the same time. In Table 1, some mechanical properties of the CNT-Cu composites and SP-CNT-Cu composites are listed. It is known that the UTS of CNT-Cu composites and SP-CNT-Cu composites is improved compared to the unreinforced Cu sample. Another point to note is that the UTS of SP-CNT-Cu composites prepared by the synergetic method of SP and FPM is much higher than that of CNT-Cu composites prepared by using only FPM. This indicates that SP can further increase the UTS of the CNT-Cu composite. 

Figure 7 displays the fracture surfaces of composites with different mass fractions of CNTs. SEM images in Figure 7a,c,e depict fracture morphologies for 0.1 wt %, 0.3 wt %, and 0.5 wt % CNT-Cu composites, respectively. Figure 7b,d,f present fracture morphologies for SP-0.1 wt %, SP-0.3 wt %, and SP-0.5 wt % CNT-Cu composites. In Figure 7a–d, dimples, which are typical characteristics of plastic fracture, can be observed. The pull-out and broken CNTs indicate that load transfer plays a major role in the stretching process. For the SP-0.5 wt % CNT-Cu and 0.5 wt % CNT-Cu composites, it can be found in Figure 7e–f that CNT clusters exist in the fracture surface. This means that the mass fraction of CNTs that reached 0.5 % in the matrix would lead to defects (voids or pores), following the lower strengthening effect and result in early initiation of cracks at defects.

### 3.4. Interface Characteristics

With the tensile strength of the composites prepared in this work, when the mass fraction of CNTs is 0.1% or 0.3%, the tensile strength of the composite is increased by the addition of CNTs. According to the literature, the strengthening mechanism of CNTs on metal matrix composites (MMCs) exhibits grain refinement [28], load transfer [29], thermal mismatch, Orowan mechanisms [30], etc. However, it is well-known that the greatest contribution to the improvement of tensile strength is the load transfer. The load transfer efficiency depends on the aspect ratio of the CNTs and the bonding strength between the CNTs and the matrix. Therefore, in order to investigate the reason that the tensile strength of the SP-CNT-Cu composite is higher than that of the CNT-Cu composites when the mass fraction of the CNTs is 0.3%, the interface of the composites must be characterized by TEM observation.

Figure 8a shows the TEM microstructure of the 0.3 wt % CNT-Cu composite prepared by only FPM. It can be observed that there is bond between the CNTs and the Cu matrix without interfacial gaps. The CNT-Cu interface could be divided into three representative regions in terms of the features reflected by the TEM images taken at the different interface regions of B, C, and D, as marked in Figure 8a. 

Region B has been analyzed by fast Fourier transform (FFT) and inverse fast Fourier transform (IFFT). In region B, presented by Figure 8b, the FFT shows the (111) and (200) characteristic diffraction spots of Cu_2_O, in addition to the (111) characteristic diffraction spots of Cu. The angle between diffraction spots (111) _Cu2O_ and (200)_Cu2O_ is 54°, the angle of measured value, and calculated value are in good agreement. According to the noise-filtered IFFT image, the lattice inter-planar spacing is measured to be 0.237 nm, well matching the displacing of the (111)_Cu2O_ plane (the standard values of the lattice inter-planar spacing of (111)_Cu2O_ is 0.246 nm, as provided in the PDJCS data) [31]. Additionally, the same FFT/IFFT results can be obtained in regions C and D. The FFT/IFFT patterns, as shown in Figure 8c,d respectively, confirm the presence of (111) _Cu2O_ and (200) _Cu2O_ planes in regions C and D. Thus, the Cu_2_O phase between the Cu matrix and the CNTs is confirmed, suggesting that interfacial oxygen atoms occurred during the fabrication process. On the other hand, it could be observed from the IFFT of region B (Figure 8b) that there is a presence of high density of dislocation areas [32] in the selected region, as marked with yellow words “distortion area”. The high density of dislocations is also presented in the distortion areas of regions C and D, as shown in Figure 8c,d. 

Figure 9a shows the TEM image of the SP-0.3 wt % CNT-Cu composite. It can be seen clearly that nanoparticles are present on the surface of the CNTs exposed from the Cu matrix. Figure 9b displays the HRTEM image of this sample. Similarly to Figure 8a, four representative regions were selected for analysis and as marked C, D, E, and F. For region C, the matrix does not have direct contact with the CNTs observed from Figure 9b. The FFT displayed in Figure 9c shows the characteristic (111) and (-220) diffraction spots of Cu, and the characteristic (111) and (311) diffraction spots of Cu_2_O. In addition, the lattice inter-planar spacing of (111)_Cu_ and (111)_Cu2O_ are measured to be 0.211 nm and 0.241 nm, respectively, in terms of the noise-filtered IFFT image, best matching the displacing of the (111)_Cu_ and (111)_Cu2O_ plane (the standard values of the lattice inter-planar spacing of (111)_Cu_ and (111)_Cu2O_ are 0.209 nm and 0.246 nm, respectively, as provided in the PDJCS data). This result is the same as the result of Figure 8, the interfacial oxygen atoms exist between the Cu matrix and the CNTs. In order to explore the nanoparticles at the interface, the FFT/IFFT of regions D, E, and F are displayed as Figure 9d–f, respectively. Figure 9d shows that region D has only (200) _Cu2O_ diffraction spots, which means that Cu_2_O nanoparticles exist in this region, and that there are a lot of dislocations in the bottom right part of this area. In addition, Figure 9e,f indicate that the middle region of E and F are both Cu nanoparticles. According to Figure 9d–f, it has been confirmed that these nanoparticles are Cu particles and Cu_2_O particles. Their particle size in the SP-CNT-Cu composite is significantly reduced compared with that in the CNT-Cu composite powder. It is presumed that the particles are worn down by the electron beam while the sample is thinned.

From the characterization results of TEM, both the 0.3 wt % CNT-Cu composite and the SP-0.3 wt % CNT-Cu composite has interfacial oxygen atoms at the interface between the CNTs and the Cu matrix. Moreover, there are distortion regions with high dislocation densities near the interface. Both of them are beneficial to improve the load transfer efficiency of the CNTs during the stretching process. The interfacial oxygen atoms of the SP-0.3 wt % CNT-Cu composite could stem from two aspects: one is oxygen-containing functional groups on the surface of CNTs, and the other is Cu_2_O nanoparticles that have not been completely reduced, which is consistent with the XRD pattern of Figure 3. As for the 0.3 wt % CNT-Cu composite, the source of the oxygen atoms could only be the oxygen-containing functional group on the surface of the CNTs. The interfacial oxygen atoms play an important role in enhancing the strength of CNT-Cu composites. It could provide chemical interfacial bonding between Cu matrices and CNTs [31]. Additionally, the presence of a high density of dislocations implies the existence of internal stress, which provides a good mechanically interfacial bonding at the interface. The feature of high density of dislocations is attributed to the mismatch in the coefficient of thermal expansion (CTE) of these different phases, resulting in the lattice structure distortion near the interface. Therefore, the prominent internal stresses at these distortion [33,34] areas lead to mechanical interfacial bonding.

The above discussion illustrates the improvement in load transfer efficiency by chemical bonding and mechanical bonding at the interface. The reason that the tensile strength of the SP-CNT-Cu composite is higher than that of the CNT-Cu composite when the mass fraction of the CNTs is 0.3% must be further discussed. On the one hand, the oxygen content at the interface is worthy of attention. According to our previous report [35], as the CNT-CuAc precursor solution is subjected to SP, the R (I_D_/I_G_) value of the CNTs of the Roman spectrum is reduced compared to the original CNTs. This indicates that the oxygen-containing functional groups on the surface of the carbon nanotubes must be reduced. However, after SP, a small amount of Cu_2_O particles attach to the surface of the CNTs, which may increase the oxygen content of the CNTs. On the other hand, for SP-CNT-Cu composites, the Cu nanoparticles and Cu_2_O nanoparticles at the interface must affect the tensile strength. The 0.3 wt % CNT-Cu composite has no nanoparticles on the surface of its CNTs, as shown Figure 8. Its strength is also lower than that of the SP-0.3 wt % CNT-Cu composite, indicating that these nanoparticles prepared by SP could increase the strength further. Interface diagrams of a CNT-Cu composite and an SP-CNT-Cu composite are shown in Figure 10. These nanoparticles not only act as rivets between the CNTs and Cu matrix to improve interface bonding, but also hinder the movement of dislocations, which is also known as the Orowan strengthening mechanism [36,37,38]. While the dislocations encounter these nanoparticles, these nanoparticles would block the movement of dislocations, thus improving the strength of the composite.

## 4. Conclusions

The SP process is an effective technological approach to modifying the surface of CNTs, and the adhesion of nanoparticles on the surface of CNTs is beneficial for improving the dispersion of CNTs. When the mass fraction of CNTs is less than 0.5%, the tensile strength of the composites increased with the additional CNT content. At the interface between the CNTs and the Cu matrix, interface oxygen atoms and internal stress provide chemical bonding and mechanical bonding, respectively, for the interface, which is beneficial for the load transfer efficiency of the CNTs. Under the same preparation conditions, the tensile strength of the SP-CNT-Cu composites is greater than that of the CNT-Cu composites, because of the nanoparticles formed on the surface of CNTs.

## Figures and Tables

**Figure 1 materials-12-00670-f001:**
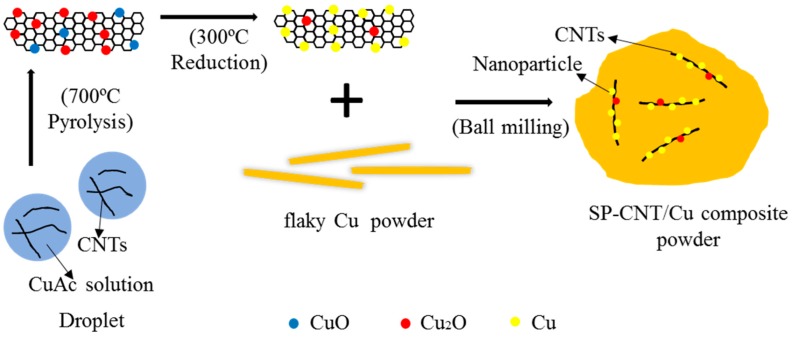
Illustration of fabrication procedure of CNT-Cu composite powder.

**Figure 2 materials-12-00670-f002:**
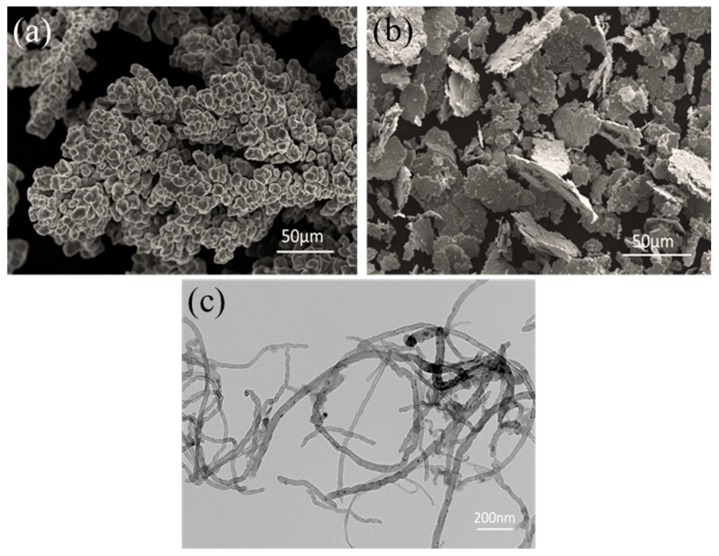
(**a**) SEM images of the electrolytic copper powder; (**b**) SEM image of the flaky Cu powder; (**c**) TEM image of the CNTs.

**Figure 3 materials-12-00670-f003:**
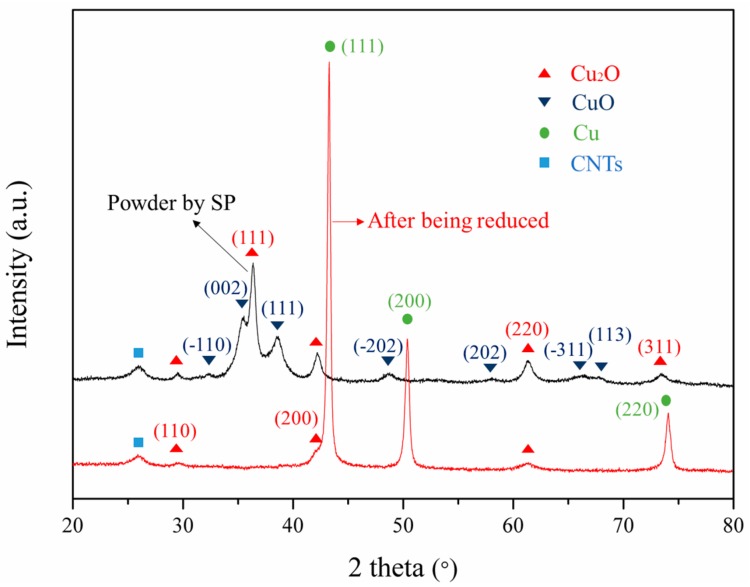
XRD patterns of the obtained composite powder by SP and after being reduced under an H_2_ atmosphere.

**Figure 4 materials-12-00670-f004:**
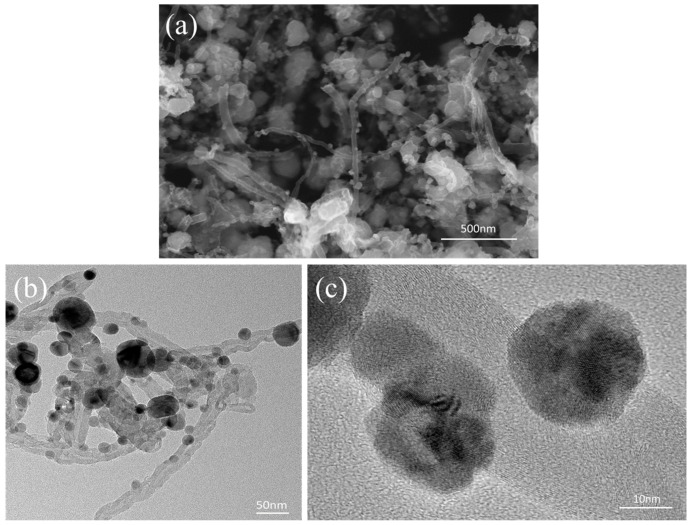
(**a**) SEM image of the CNT-Cu composite powder; (**b**) TEM image and (**c**) HRTEM image of the CNTs-Cu composite powder.

**Figure 5 materials-12-00670-f005:**
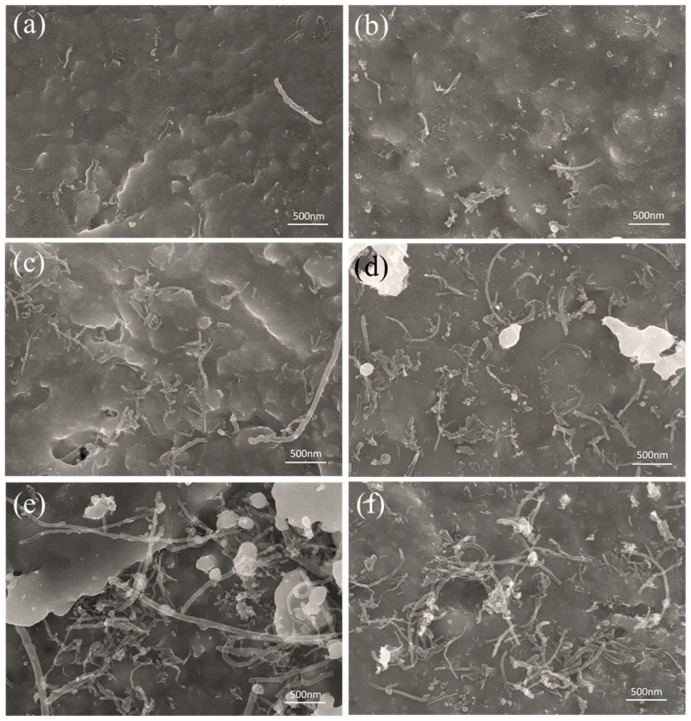
(**a**,**c**,**e**) SEM images of the 0.1 wt %, 0.3 wt %, and 0.5 wt % CNT-Cu composite powder, respectively; (**b**,**d**,**f**) SEM images of the SP-0.1 wt %, SP-0.3 wt %, and SP-0.5 wt % CNT-Cu composite powder, respectively.

**Figure 6 materials-12-00670-f006:**
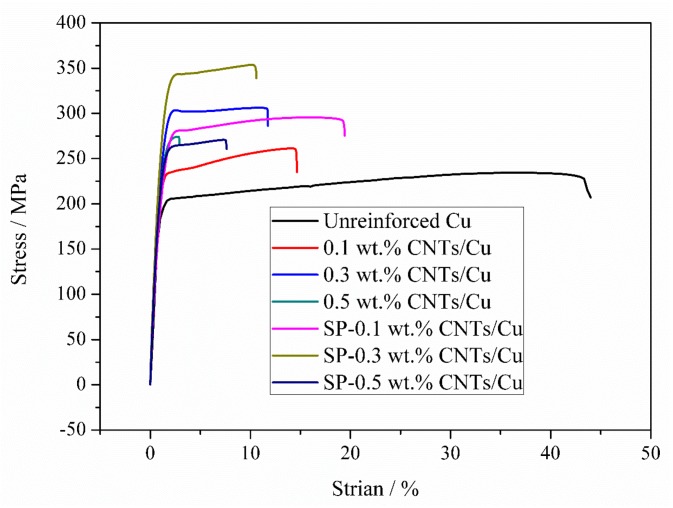
Stress-strain curves of unreinforced Cu, CNT-Cu composites and SP-CNT-Cu composites with different CNT content, respectively.

**Figure 7 materials-12-00670-f007:**
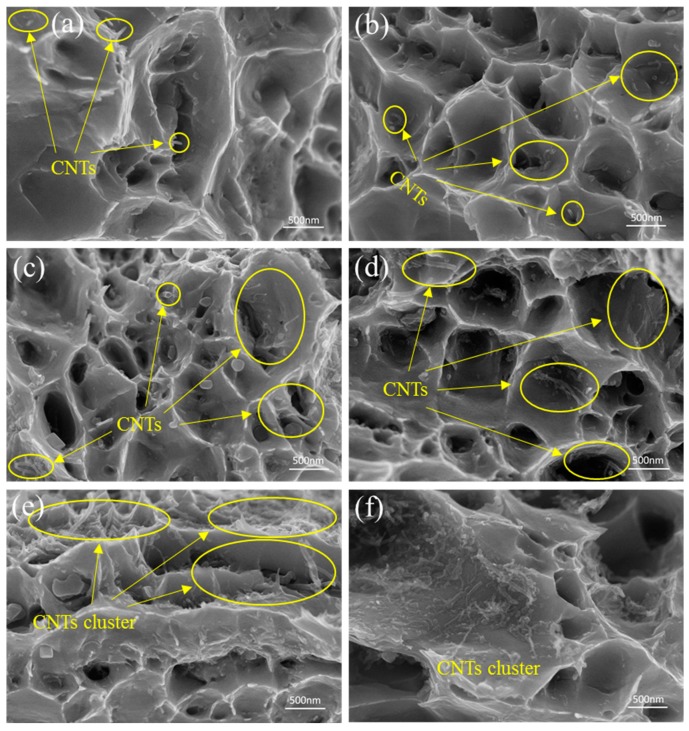
(**a**,**c**,**e**) SEM images of fracture morphology for 0.1 wt %, 0.3 wt %, and 0.5 wt %; (**b**,**d**,**f**) SP-0.1 wt %, SP-0.3 wt %, and SP-0.5 wt % CNT-Cu composites, respectively.

**Figure 8 materials-12-00670-f008:**
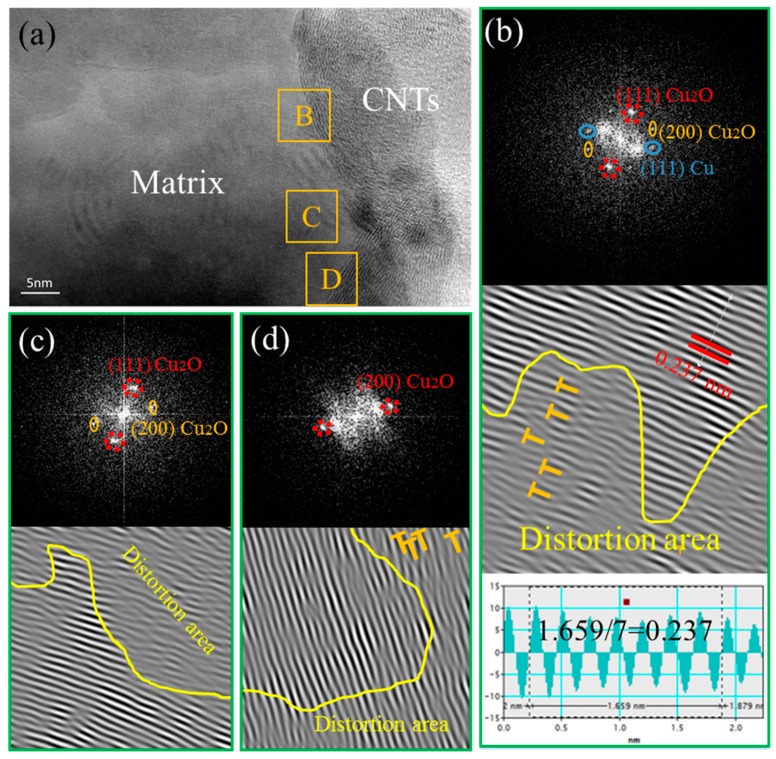
(**a**) TEM image of 0.3 wt % CNT-Cu composite; (**b**–**d**) fast Fournier transform (FFT) (upper) and inverse fast Fournier transform (IFFT) (under) of regions B, C, and D, respectively.

**Figure 9 materials-12-00670-f009:**
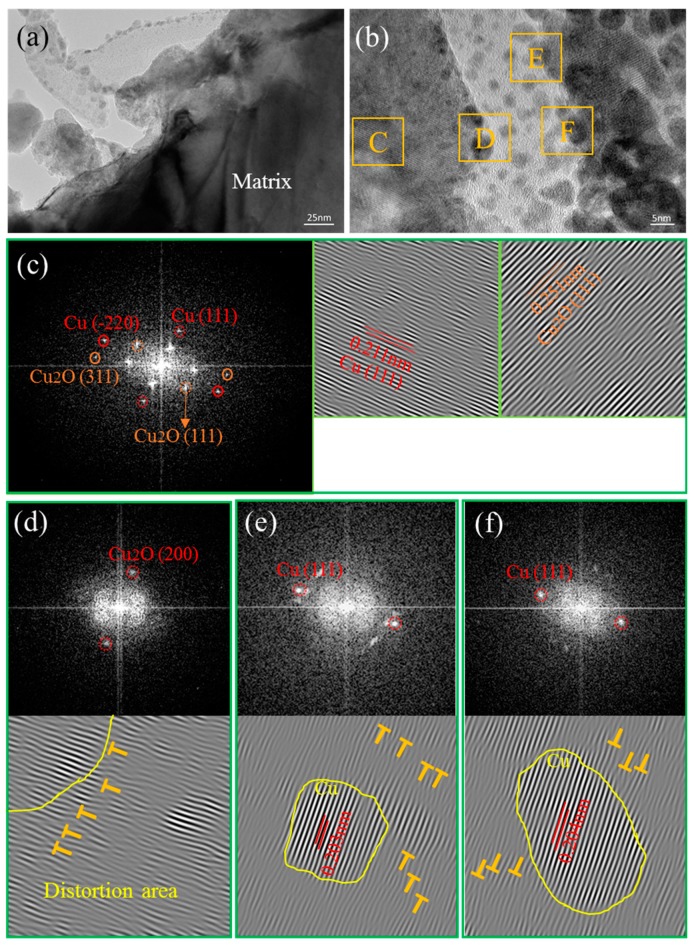
(**a**,**b**) TEM image of SP-0.3 wt % CNT-Cu composite; (**c**–**f**) FFT and IFFT of regions C, D, E, and F, respectively.

**Figure 10 materials-12-00670-f010:**
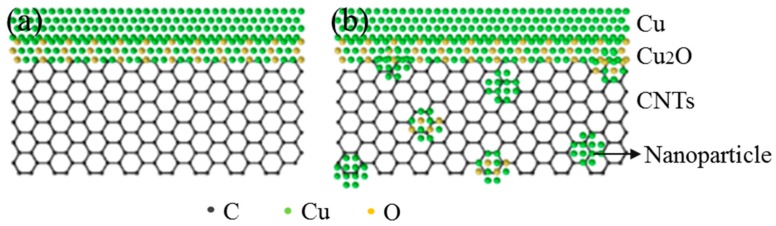
(**a**) Interface diagram of CNT-Cu composite; (**b**) Interface diagram of SP-CNT-Cu composite.

**Table 1 materials-12-00670-t001:** Comparison of tensile properties of composites prepared in this work.

Sample	Relative Density (%)	Hardness (HV)	UTS (MPa)	Elongation (%)	Enhancement (%)
Unreinforced Cu	98.7	68.5	234.2 ± 6.0	43.3 ± 0.3	--
0.1 wt % CNT-Cu	98.6	87.1	261.2 ± 5.5	14.6 ± 0.3	11.5
0.3 wt % CNT-Cu	97.7	95.3	306.2 ± 4.4	11.7 ± 0.5	30.7
0.5 wt % CNT-Cu	96.1	98.8	274.1 ± 3.5	2.9 ± 1.0	17.1
SP-0.1 wt % CNT-Cu	98.7	94.4	295.5 ± 4.8	19.4 ± 0.5	26.2
SP-0.3 wt % CNT-Cu	97.9	102.3	353.9 ± 4.8	10.6 ± 0.5	51.5
SP-0.5 wt % CNT-Cu	97.7	105.65	270.7 ± 5.0	7.6 ± 0.6	15.6

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
