# Peer review of "Enhancing Interfacial Bonding and Tensile Strength in CNT-Cu Composites by a Synergetic Method of Spraying Pyrolysis and Flake Powder Metallurgy"

_materials, 2019, doi:10.3390/ma12040670_

Round 1
Reviewer 1 Report
This paper entitle “ Enhancing interfacial bonding and tensile strength in CNTs/Cu composites by a synergetic method of spraying pyrolysis and flake powder metallurgy” provides another useful method combining two well-known strategies for CNTs/Cu composites. This paper is well written and shows good characterization results and conclusions. However, some improvements need to be performed:
1. The reference “Journal of Alloys and Compounds, 47, 91-99 (2018) (https://doi.org/10.1016/j.jallcom.2018.03.029)” should be cited and commented in the introduction section since it deals with the fabrication of CNT/Cu composites with enhanced strength and ductility by the same methods. Therefore, the authors should remark the novelty of this paper with respect to this already published paper because now its novelty is under question.
2. The authors should mention in the introduction section why they chose CNTs as reinforcing agents in comparison with other reinforcing carbon nanomaterials.
Author Response
Dear Editors,
Thank you for sending us the comments on our manuscript (Materails-435807). We very appreciated the comments and suggestions which are very helpful for improving the quality of our manuscript. We have seriously addressed these comments in this revision.
Reviewer #1: This paper entitle “Enhancing interfacial bonding and tensile strength in CNTs/Cu composites by a synergetic method of spraying pyrolysis and flake powder metallurgy” provides another useful method combining two well-known strategies for CNTs/Cu composites. This paper is well written and shows good characterization results and conclusions. However, some improvements need to be performed:
1.The reference “ Journal of Alloys and Compounds, 47, 91-99 (2018) should be cited and commented in the introduction section since it deals with the fabrication of CNTs/Cu composites with enhanced strength and ductility by the same methods. Therefore, the authors should remark the novelty in under question.
Response:
Thanks for your kind comments. It has been modified in the revised manuscript.
2. The authors should mention in the introduction section why they chose CNTs as reinforcing agents in comparison with other reinforcing carbon nanomaterials.
Response:
Thanks for your kind comments. It has been modified in the revised manuscript.
I regretted that there were some grammatical errors in the manuscript, I had checked the paper again and carefully proof-read the manuscript.
At last, we are grateful to the reviewers for the careful revision of this paper, and the reviewers' comments have given us a better understanding of the paper and its related knowledge. Thus, at the end of the letter, we would sincerely thank the reviewers and editors once again.
Sincerely yours,
All authors.

Reviewer 2 Report
File attached with comments. Overall, good paper that will yield further areas of growth with CNTs.

Author Response
Dear Editors,
Thank you for sending us the comments on our manuscript (Materails-435807). We very appreciated the comments and suggestions which are very helpful for improving the quality of our manuscript. We have seriously addressed these comments in this revision.
Reviewer #2
1. Introduction
Paragraph 4 – Please highlight FPM – what is it? You also previously discuss need for chemical compatibility of CNT with Cu, then make a comment about geometric compatibility. Why change in focus? Is SP=SPS? Please clarify.
Response: It has been corrected in the revised manuscript. FPM = flake powder metallurgy. SP = spraying pyrolysis. SPS = spark plasma sintering. In this work, we use a synergetic method of spaying pyrolysis (SP) and flake powder metallurgy (FPM) to prepare composite material. The chemical compatibility is discussed because SP method was employed at first to improve the wettability between carbon nanotubes (CNTs) and copper (Cu) by achieving chemical bonding at the interface. After that, FPM process was introduced. Besides, according to the reports of Jiang [1] and Fan [2] et al., the dispersion capacity of CNTs was greatly enhanced by changing spherical powder into disc-shaped flakes. Therefore, we mentioned geometric compatibility in the manuscript.
[1] Fan, G.; Jiang, Y.; Tan, Z.; Guo, Q.; Xiong, D.; Su, Y.; Lin, R.; Hu, L.; Li, Z.; Zhang, D., Enhanced interfacial bonding and mechanical properties in CNT/Al composites fabricated by flake powder metallurgy. Carbon 2018, 130, 333-339.
[2] Jiang, L.; Fan, G.; Li, Z.; Kai, X.; Zhang, D.; Chen, Z.; Humphries, S.; Heness, G.; Yeung, W. Y., An approach to the uniform dispersion of a high volume fraction of carbon nanotubes in aluminum powder. Carbon 2011, 49, (6), 1965-1971.
In the last sentence, you state unreinforced Cu bulk and CNTs/Cu composites as control – is CNT/Cu the control or the experiment. Please clarify.
Response: In this work, a synergetic method of SP and FPM was used to prepare SP-CNTs/Cu composite as an experimental group. The unreinforced Cu bulk and CNTs/Cu composites as control groups were established by FPM only.
2. Experiment
You may want an introductory statement as to your general procedure, then go into detail. Why do you start with an oxide?
Response: In the experimental part, we described it according to the experimental process. After the copper acetate (CuAc) solution is spray pyrolyzed, it is thermally decomposed into copper oxide and cuprous oxide in terms of the subsequent XRD pattern. Therefore, after the SP of the CNTs and CuAc mixture, the obtained reaction product is CNTs-Cu oxide composite powder.
Please fix the technical writing in this section. Do not start sentences with numbers.
Response: Thanks for mentioning this error. It has been corrected in the revised manuscript.
In section 2.2, it would be advantageous to show the chemical reaction.
Response: Thanks for your comments. This is a hydrogen reduction of copper oxide, cuprous oxide reaction. It would be added in the revised manuscript.
How do you verify if you add 0.1CNT, that you actually get 0.1CNT in the final product?
Response: Based on previous experience, the CNTs-CuAc precursor was composed of 9.0 grams of CNTs dispersion, 6.0 grams of copper acetate monohydrate, and 3.0 liters of deionized water. After spray pyrolysis and reduction of the precursor, the carbon content of the obtained product was detected in a C/S analyzer (ELTRA CS800). In this work, the mass fraction of carbon in the composite powder is 15.89%. Subsequently, these composite powders would be ball milled with flake copper powder in a certain ratio to ensure that the mass fraction of CNTs in the final product is 0.1%, 0.3% and 0.5%, respectively.
Section 2.3 – for the hardness/mechanical testing, are you following a standard?
Response: For these test parameters, I refer to the following literature:
[1] Yang, Z. Y.; Wang, L, D.; Shi, Z. D.; Wang, M., Preparation mechanism of hierarchical layered structure of grapheme/copper composite with ultrahigh tensile strength. Carbon 2018, 127, 329-339.
[2] Liu, L.; Bao, R.; Yi, J. H.; Fang, D., Fabrication of CNT/Cu composites with enhanced strength and ductility by SP combined with optimized SPS method. Journal of Alloys and Compounds 2018, 747, 91-99.
[3] Liu, L.; Bao, R.; Yi, J. H.; Li, C. J., Well-dispersion of CNTs and enhanced mechanical properties in CNTs/Cu-Ti composites fabricated by Molecular Level Mixing. Journal of Alloys and Compounds 2017, 726, 81-87.
3. Results and discussion
Paragraph 1 – Is there supporting documentation that there is a greater surface area, and this is the advantage for the CNT? Or can it be the oxide reaction? Or vanderwaals?
Response: It has been corrected in the revised manuscript. There is no relevant supporting literature for it with a greater surface area, but we can find supporting documentation [1] [2] that the dispersion capacity of CNTs was greatly enhanced by changing spherical powder into disc-shaped flakes.
[1] Jiang, L.; Fan, G.; Li, Z.; Kai, X.; Zhang, D.; Chen, Z.; Humphries, S.; Heness, G.; Yeung, W. Y., An approach to the uniform dispersion of a high volume fraction of carbon nanotubes in aluminum powder. Carbon 2011, 49, (6), 1965-1971.
[2] Fan, G.; Jiang, Y.; Tan, Z.; Guo, Q.; Xiong, D.-b.; Su, Y.; Lin, R.; Hu, L.; Li, Z.; Zhang, D., Enhanced interfacial bonding and mechanical properties in CNT/Al composites fabricated by flake powder metallurgy. Carbon 2018, 130, 333-339.
Section 3.2 – can you confirm that the reduction does not react with the CNTs?
Response: It is difficult to confirm that the reduction dose not react with the CNTs from the XRD pattern. However, there is a related literature [1] mentioning that the oxygen-containing function groups on the CNTs are difficult to be reduced at the reduction temperature in this paper.
[1]Kim, K. T.; Cha, S. I.; Gemming, T., The role of interfacial oxygen atoms in the enhanced mechanical properties of carbon-nanotube-reinforced metal matrix nanocomposites. Small 2008, 11, 40-46.
“its characteristic peaks is not very noticeable because the crystallinity of CNTs is not pretty well”. Please clarify this statement – the intention is not clear.
Response: Thank you for pointing out this unclear sentence. It would be modified in the revised manuscript. In this paper, the characteristic peaks of CNTs are not noticeable, because carbon content of the composite powder for XRD characterization is 15.89%. And, the intensity of the characteristic peak of CNTs is low due to the low mass fraction. In addition, the distortion of the crystal lattice of the CNTs is also serious, which leads to the low characteristic peak intensity.
For Fig 4 – how would you prefer the dispersion? Single threads or knots?
Response: Single threads dispersion of CNTs are more conducive to improving the mechanical properties of the composite compared with the entangled ones. Hence, we prefer the single threads of CNTs in reinforcing the metal matrix composites.
“It is beneficial to retain the integrity of the CNTs and ensure the nanoparticle Cu/Cu2O is attached to the CNTs. The interfacial…” The sentence “Copper nanoparticles with size range…” is a very long sentence. Please clarify. How do the Cu nanoparticles help impede aggregation?
Response: CNTs-Cu composite powder can be dispersed uniformly in the Cu matrix powder due to their relatively decentralized structure with a moderate ball milling process, it is helpful to retain the structure that nanoparticles attached to CNTs and the integrity of the CNTs. TEM and HRTEM are used to investigate the interfacial structure between CNTs and nanoparticles. Nanoparticles with the size range from 20 to 50 nm are uniformly attached to the surface of the CNTs as shown in Fig. 4(b), and the Fig. 4(c) shows that the nanoparticles are attached uniformly to the CNTs, these attached nanoparticles could further impede the aggregation of CNTs.
Figure 4 shows that some nanoparticles attached to the surface of CNTs. These nanoparticles act a modifiable role in the CNTs, which is beneficial to improve the dispersion of CNTs in the matrix copper powder. Thank you for your reminder, this inappropriate expression would be modified in the revised manuscript.
Further on, when discussing Fig 5 – you state there is crushing/welding. There is no welding here – consider using the terminology plasticity or ductility
Response: Thanks for mentioning this error. It has been corrected in the revised manuscript.
You state the bundles have a negative effect on the mechanical properties – do you have supporting evidence for that conclusion? Were you able to confirm that if you put in 0.1CNT that you got 0.1CNT?
Response: For the carbon nanomaterial reinforced metal matrix composites, the mechanical strength of the composite will be decreased when the mass fraction of CNTs exceeds a certain amount, which is caused by the agglomeration of the CNTs. In this paper, the subsequent tensile strength indicates that when the mass fraction of CNTs is 0.5%, the tensile strength of the composite does decrease evidently. Therefore, it is considered that the CNTs bundles in Fig. 5 has a negative impact on the mechanical properties.
After the spray pyrolysis and reduction of CNTs-CuAc precursor, the carbon content of the obtained product was detected in a C/S analyzer (ELTRA CS800). In this work, the mass fraction of carbon in the composite powder is 15.89%. Subsequently, these composite powders would be ball milled with flake copper powder in a certain ratio to ensure that the mass fraction of CNTs in the final product is 0.1%, 0.3% and 0.5%, respectively.
Section 3.4 – What is Hereon?
Response: Hereon=Therefrom.
Paragraph 2 – there is good bonding, not well bonded between CNTs and Cu.
Response: Thanks for mentioning this error. This inappropriate expression would be modified in the revised manuscript.
From FFT/IFFT patterns – where does the oxygen go and evolution of oxygen? The CNTs are not on the TEM FFT.
Response: Oxygen forms a transition layer with a few nanometers to several tens of nanometers thick at the interface between the CNTs and copper.
Lattice distortion of the CNTs is large, and diffraction spots cannot be formed. In the TEM FFT of Fig. 9, however, the diffraction ring of the CNTs can be observed.
Can you discuss why dislocations in B are heavily concentrated in the matrix, but in C and D they are with the CNTs?
Response: From the characterization of the TEM, the interface structure is: matrix copper - cuprous oxide transition - CNTs. The difference in lattice constant and coefficient of thermal expansion between copper and cuprous oxide leads to an increase in dislocations, as well as between cuprous oxide and CNTs. Besides, the thickness of the cuprous oxide layer may not be completely uniform. The dislocation of the B region is caused by the internal stress between copper and cuprous oxide, it is much closer to the matrix. The dislocations in the C and D regions are caused by the internal stress between cuprous oxide and CNTs, which are closer to the CNTs.
Paragraph 5 – last sentence “make a mechanically interfacial bonding to enhance the efficiency of load transferring”. Please clarify.
Response: A high density of dislocations is presented in the distortion area as shown in Fig. 9. The feature of this interface suggests a strong mechanically interfacial bonding has been formed between copper and CNTs. Higher interface bonding strength is beneficial to improve the load transfer efficiency when the CNTs/Cu composite undergoes the stress-strain process.
Be consistent with the location of the oxygen molecules – are they on the CNT or the Cu2O?
Response: For the SP-CNTs/Cu composites prepared by a synergetic method of spaying pyrolysis (SP) and flake powder metallurgy (FPM), oxygen should be derived from CNTs and cuprous oxide nanoparticles. However, oxygen of the CNTs/Cu composites prepared by FPM was derived from CNTs only.
I regretted that there were some grammatical errors in the manuscript, I had checked the paper again and carefully proof-read the manuscript.
At last, we are grateful to the reviewers for the careful revision of this paper, and the reviewers' comments have given us a better understanding of the paper and its related knowledge. Thus, at the end of the letter, we would sincerely thank the reviewers and editors once again.
Sincerely yours,
All authors.

Reviewer 3 Report
The paper provides a new method for improving the mechanical properties of carbon nanotube-reinforced MMCs.
The overall quality of the research and conclusions is high. The only suggestion I have is for Fig. 6. I think the two graphs (a) and (b) can be combined to allow for a better comparison between the composites made using the developed method and the ones made using the previous methods.
Author Response
Dear Editors,
Thank you for sending us the comments on our manuscript (Materails-435807). We very appreciated the comments and suggestions which are very helpful for improving the quality of our manuscript. We have seriously addressed these comments in this revision.
Reviewer #3: The paper provides a new method for improving the mechanical properties of carbon nanotubes-reinforced MMCs.
The overall quality of the research and conclusions is high. The only suggestion I have is for Fig. 6. I think the two graphs (a) and (b) can be combined to allow for a better comparison between the composites made using the developed method and the ones made using the previous methods.
Response:
Thanks for your kind comments. It has been modified in the revised manuscript.
I regretted that there were some grammatical errors in the manuscript, I had checked the paper again and carefully proof-read the manuscript.
At last, we are grateful to the reviewers for the careful revision of this paper, and the reviewers' comments have given us a better understanding of the paper and its related knowledge. Thus, at the end of the letter, we would sincerely thank the reviewers and editors once again.
Sincerely yours,
All authors.

Round 2
Reviewer 1 Report
The paper is much better explained now and deserves publication after a minor revision:
I suggest to enhance the writing and information of the new added sentence “CNTs have the valuable properties compared with other carbon nanomaterials, such as superior strength(up to ~100 GPa), superior elastic modulus(~1 TPa) and highaspect ratio(>1000)[5-7]” as follows:
“CNTs shows excellent strength(up to ~100 GPa) and elastic modulus(~1 TPa), and high aspect ratio(>1000)[5-7] like other carbon nanomaterials such as graphene oxide [Scientific Reports 7:11684 (2017). http://doi.org/10.1038/s41598-017-10260-x]
Author Response
Dear Editors,
Thank you for sending us the comments on our manuscript (Materails-435807). We very appreciated the comments and suggestions which are very helpful for improving the quality of our manuscript. We have seriously addressed these comments in this revision.
Reviewer: The paper is much better explained now and deserves publication after a minor revision:
I suggest to enhance the writing and information of the new added sentence “CNTs have the valuable properties compared with other carbon nanomaterials, such as superior strength (up to ~100 GPa), superior elastic modulus (~1 TPa) and high aspect ratio (>1000) [5-7]” as follows:
“CNTs shows excellent strength (up to 100 GPa) and elastic modulus (~1 TPa), and high aspect (>1000) [5-7] like other carbon nanomaterials such as grapheme oxide and carbon nanofibers.”
Response: Thanks so much for your kind comments. It has been modified in the revised manuscript.
We are grateful to the reviewers for the careful revision of this paper, and the reviewers' comments have given us a better understanding of the paper and its related knowledge. Thus, at the end of the letter, we would sincerely thank the reviewers and editors once again.
Sincerely yours,
All authors.